# Design and Performance Analysis of LARMbot Torso V1

**DOI:** 10.3390/mi13091548

**Published:** 2022-09-18

**Authors:** Wenshuo Gao, Marco Ceccarelli

**Affiliations:** Laboratory of Robot Mechatronics, University of Rome Tor Vergata, 00133 Rome, Italy

**Keywords:** humanoid robots, humanoid torso, experimental characterizations, cable-driven mechanism, experimental testing, LARMbot humanoid, LARMbot torso

## Abstract

In this paper, laboratory experiments of LARMbot torso V1 are reported in the third mode, thereby providing a testing characterization. Sensors were used to measure parameters including the contact force between the shoulder and cables, linear acceleration, angles of the torso body, and power consumption. The results showed that the LARMbot torso V1 can bend successfully to the desired angles, and that it is able to complete a full motion smoothly. The LARMbot torso V1 can mimic human-like motiaons. Based on our analysis of the test results, improvements are suggested, and new designs are considered.

## 1. Introduction

Humanoid robots are more and more popular in our daily life. They are used for public services and to provide assistance to people at home. The humanoid torso, as the main part of a humanoid robot, is critical to achieving human-like motions.

In recent years, novel mechanisms for humanoid torsos have been proposed by scholars in order to obtain better performance of humanoid robots. Sometimes, torsos with box shapes are used to support and connect with limbs and heads, e.g., the MIT humanoid robot [1] and the Quori robot [2]. Seiwald et al. [3] proposed a LOLA v1.1 humanoid robot that has independent pelvis rotation and adduction between the torso and legs, with a motion that is driven by motors with limited movements. Cao et al. [4] developed a 2-dof torso with two identical joints and a series of mechanisms. Sander et al. [5] designed a torso joint for the ARMAR humanoid robot that can contribute to the pitch and roll movements. These box-shaped torsos are not lightweight enough and do not contribute to improving the structures. Ranjans et al. [6] proposed a humanoid robot named AUTOMI with a T-shape torso. A lightweight, articulated torso with universal joints was designed to be manufactured by a 3D printer for the AUTOMI robot to help achieve proper motion [7]. However, dew scholars have focused on parallel kinematic mechanisms (PKM) and used them for torso design in humanoid robots. Huang et al. [8] proposed a 3-dof PKM torso based on analyses of the structures of humans and animals. Fiorio et al. [9] proposed a torso based on the parallel kinematic mechanism; the majority of its joints were designed with serial or differential mechanisms, which led to the use of less parallel kinematic structures, thereby decreasing the complexity. However, most of the parts were made of metal, which increased the weight of the design. Boblan et al. [10] proposed a humanoid torso named ZAR5. It was actuated by artificial air muscles with human-like proportions and functionality. Reinecke et al. [11] proposed several torsos, including an anthropomorphic structure and a torso joint with an adjustable linear spring mechanism. Pencic et al. [12] proposed a multi-joint structure with a stiff, low backlash and a self-locking mechanism with small actuators including sets of gears to perform motions. These solutions can increase the flexibility of the humanoid robots and contribute to the emulation of human-like motions. However, the structures of these torsos are complicated and are not of high modularity. In addition, these torsos are heavy because of the presence of metallic parts. Alamag et al. [13] tested a hybrid manipulator humanoid torso that was made of joints and 3D printed components. It could easily mimic the torso movements of a human in terms of bending angles for roll, pitch, and yaw. Li et al. [14] proposed a humanoid torso driven by disk-type actuators with a 6 -dof serial mechanism that was similar to the human spine both in shape and features. However, it was not applied to a humanoid robot. Osada et al. [15] designed a multiple spine structure with a planar-muscle-driven mechanism as a humanoid trunk, which can effectively perform human-like motions; however, the mechanism requires a large workspace for its operation. Recently, several versions of humanoid torsos were developed by the LARM team, e.g., the CaPaMan-based trunk design in 2001, the CALUMA design in 2003, the PKM design in 2008, the waist-trunk system using two CaPaMan manipulators in 2010 [16], the waist-trunk system with CaPaMan 2bis structure, and the trunk-waist design with two PKM modules, as reported in [17]. Cafolla et al. [18] proposed a humanoid torso (LARMbot torso V1) made of three vertebral-discs-joint units. Experiments were conducted to verify that the proposed design could properly reproduce human-like motions [19].

In this paper, lab experiments are discussed based on test results to characterize the LARMbot torso V1. The paper examines the mechanical design and operation of the humanoid torso, sized at a reduced scale, with biomimetic features and functions. An inertial measurement unit (IMU) sensor was applied to test the linear acceleration and angle positions during testing of the motions. Force sensors were used to measure the contact forces between the shoulder and cables to evaluate the tension of the cable actuators. Power consumption was determined using a current sensor. After presenting our analysis, improvements are suggested for future designs of the LARMbot torso. Two possible schemes are proposed for a new humanoid torso on the basis of our characterization of the current prototype.

This paper is organized as follows: Section 2 describes the requirements for designing a humanoid torso; Section 3 introduces the vertebra-disc unit mechanism and the prototype of the LARMbot torso V1; Section 4 describes the test setup and modes; Section 5 provides the results and a discussion; Section 6 presents considerations regarding the problems of the LARMbot torso V1 and proposes new designs for the vertebra-disc units; finally, Section 7 summarizes the contributions of this paper.

## 2. Requirements for a Humanoid Torso

It is important to design a humanoid torso with anthropomorphic characteristics, because this is useful for the recreation of human-like motions. The human torso consists of the thorax, spine, pelvis, and muscles, as shown in Figure 1a [20]. In general, several organs are contained in the thorax, such as the heart, lungs, liver, and stomach, which are protected by the rib cage. For the motions of a human torso, the spine can be considered the predominant driving part, as analyzed by S. Gracovetsky [21]. The human spine consists of vertebras, intervertebral discs, anterior longitudinal ligament, posterior longitudinal ligament, and ligamenta flava. An intervertebral disc connects two vertebras and has a suitable stiffness for elastic behavior. It can decrease the impact from external loads and can increase the motion workspace of the spine. The anterior longitudinal ligament and posterior longitudinal ligament can limit the positions of the spine and intervertebral discs. Figure 1b shows a mechanical model of the composition of the vertebra–intervertebral disc unit with the related muscle complex in the human spine. The purpose of this model is to identify the essential components to replicate in the structure of a humanoid robot. Based on the aforementioned anatomical structures, the human torso can move in three modes, i.e., flexion-extension, lateral bending, and transverse rotation, as summarized in Table 1 [17,20]. As for flexion-extension motion, a standing adult can bend forward up to around 45 deg. An adult in seated configuration can bend left and right by up to around 40 deg, while 50 deg is the maximum range that a human can transversally rotate [12,20].

Based on the aforementioned characteristics, several requirements must be considered for designing a humanoid torso; the main ones are listed in Figure 2.

In general, a proper structure is a fundamental factor when designing a humanoid torso. Attention must be paid to the main aspects, e.g., size, payload, and assembly method. A humanoid torso should be of a suitable size to have a good ratio with the sizes of the other parts of the robot, such as the limbs, arms, and head. In addition, the limbs and head should be easily attachable. Moreover, the humanoid torso should be capable of supporting the payload and torque from the arms and head.

Similarly, a humanoid torso should be operated properly performing the tasks with human-like motions. The control system should be developed with proper logic also for autonomous operation. In specific assignments, users can operate the humanoid torso by using a switch or a joystick or a voice controller, or orders on a laptop. A humanoid torso should be designed with features such as capability for human-like motions, lightweight structure, low-cost assembly, and low power consumption. A humanoid torso should perform a human-like motion while supporting limbs doing their actions. In addition, a good humanoid torso is designed with proper stiffness for mimicking a human motion and supporting other parts of a humanoid robot properly. In general, a humanoid torso can move ±30 deg in pitch and roll, and ±40 deg in yaw. A humanoid torso should be manufactured in suitable ways and be with proper materials so that it is made of the lightweight structure. An affordable humanoid robot can contribute to a servicing application in human daily life. Hence, the cost of a humanoid torso must be at reasonable level not increasing the expense of the whole robot. Power consumption is also convenient at a low level for a proper operation duration also because limbs can consume a lot of energy from humanoid batteries in performing the assignments.

## 3. LARMbot Torso V1 Prototype

The above-mentioned requirements have been considered also for an experimental characterization of LARMbot torso V1. According to the structure of the human spine, a spine scheme can be modelled as in Figure 1b. It can be regarded as a unit of a series of vertebra-disc units with four cables. Based on this structure model, a kinematic scheme of a vertebra-disc unit with two cables is presented in Figure 3, referring to a vertebra-spring cable-driven mechanism [22]. The design structure consists of two cables, two actuators, two vertebras, and a coupling with spring. The intervertebral disc is modeled as a flexible coupling with a spring. The mechanism is driven by 2 servo motors that actuate the series of vertebra-disc units giving actuation to the unit in Figure 3 at actuation points A_1_ and A_2_ in Figure 3. The A_1_A_2_ segment link is the fixed vertebra and B_1_B_2_ is the moving vertebra body. Those vertebra platforms are designed for mimicking the vertebra-disc unit of the human spine. A_1_B_1_ and A_2_B_2_ are two cables that are actuated by servo motors. They can be understood as anterior longitudinal ligament, and posterior longitudinal ligament with muscle-like motion of the human anatomy in Figure 1b. A linear spring is designed as based on the properties of the human intervertebral disc on the human spine. The two actuations at the point A_1_ and A_2_ coming from the torso servomotors rotate the moving platform clockwise and counterclockwise, respectively, for an antagonist operation in relative motion of the two vertebra-links of the unit. Figure 3 is a model for a vertebra-disc unit with cable actuation where the servomotor drivers can be on the first unit of series of units so that points A_1_ and A_2_ indicate just that the actuation of the unit come from them.

Two Cartesian coordinate systems are assumed on the fixed vertebra and moving vertebra as O_1_x_1_y_1_z_1_ and O_2_x_2_y_2_z_2,_ respectively. The O_1_ and O_2_ are at the centers of the vertebras. The mechanism does not have motion in the Z direction, and therefore the analysis is conducted at the Oxy plane only. The rotation angle of the moving vertebra B_1_B_2_ is *θ* as per rotating around the Z axis. A model for force analysis at the point O_2_ is presented in Figure 3 yet with actions at point O_2_ as F_xi_, F_yi_, and M_i_ when i equal to 2. F_x2_, F_y2_, and M_2_ are the forces and torque at point O_2_ from the effect of linear spring; F_ex2_, F_ey2_, and M_e2_ are the external forces and torque at the point O_2_ as coming from payload and external actions. The external actions can be generated as F_e_ and M_e_ are due for example to interactions from the environment. T_1_ along A_1_B_1_ is the tension on cable with the length l_1_. T_2_ is the tension on A_2_B_2_, whose length is l_2_. The length m is t T_1_ is the tension on cable A_1_B_1_ with the length l_1_. T_2_ is the tension on A_2_B_2_, whose length is l_2_. The *m* distance is the size of O_1_A_1_ and O_1_A_2_. The *k* distance is the size of O_2_B_2_ and O_2_B_1_. Using the closed-vector loop model, the relative motion of the vertebra links can be expressed using position vectors **L_i_** of l_i_ as
(1)Li=PO1O2+RO2O1BiO2−PO1Ai
where the rotation matrix RO2O1 is given by $\left[\begin{array}{cc} \cos\theta & \sin\theta \\ -\sin\theta & \cos\theta \end{array}\right]$. Therefore, vectors **L**_1_ and **L**_2_ can be obtained to give the length of cables as follows
(2)$$L_{1}=\left[\begin{array}{c} x-k\cos\theta +m\\ y+k\sin\theta \end{array}\right]$$
(3)$$L_{2}=\left[\begin{array}{c} x+k\cos\theta -m\\ y-k\sin\theta \end{array}\right]$$
(4)$$l_{1}=\left\|L_{1}\right\|$$
(5)$$l_{2}=\left\|L_{2}\right\|$$

To drive the mechanism, it is important to provide suitable tension to the cables A_1_B_1_ and A_2_B_2_. Assuming gravity and friction of the moving vertebra as negligible, the static equilibrium equations can be formulated as
(6)$$\sum_{i=1}^{2}T_{i}=F_{e}$$
(7)∑i=12RO2O1BiO2×Ti=Me

Thus, the acting force and torque on vertebra B_1_B_2_ at O_2_ can be expressed as $F_{2\mathrm{etot}}=\left[\begin{array}{c} F_{x2}+F_{ex2}\\ F_{y2}+F_{ey2} \end{array}\right]$, and **M_2_**_etot_ = **M**_2_ + **M**_e2_.

Equations (1)–(6) can be used for the operation control of the relative motion between the two vertebra-disc units as driven by the cable length variation considering also the actions coming from the payload on the torso and external other interactions. The structure of the conditions for the equations as in Equations (1)–(6) gives also hints for the structure design of the vertebra-disc unit in the torso for a humanoid robot as applied in LARMbot torso V1.

According to the scheme in Figure 2, Cafolla et al. [19,23] designed a humanoid torso with a humanoid spine by combining three vertebra-spring cable-driven mechanisms, as shown in Figure 4b. The design solution consists of four cables, four servo motors, four vertebras, three vertebral joints, and supporting elements. Based on the characterization of the human torso in Figure 1, the humanoid LARMbot torso V1 was manufactured by using a 3D printer and market components, as shown in Figure 4a. The whole prototype is 154 × 113.8 × 261.3 mm. Couplings are used as springs, providing the support and torsion for the mechanism. Each servomotor drives a cable with the corresponding antagonist so that the structure works with 3-dof motion [23].

## 4. Materials and Methods

Experimental activity was planned to verify and deepen the capabilities of the LARMbot V1 torso with the current prototype available by using a new specific monitoring system and a specially defined experimental protocol. The components of the experimental system are clarified through Figure 5 and Figure 6 with the related explanatory text while the experimental protocol is summarized in the scheme of Figure 7 with comments to the figure on the procedural aspects. The tests of the experiments were carried out for the three modes described in Figure 7 with repetitions from 3 to 5 times of which the illustrative results are reported with reference to the full rotation mode which includes the other two bending modes in Figure 7.

The control system of the LARMbot V1 torso consists of Arduino boards, PC, and sensors and is used also as the testing set up for monitoring the tests as shown in Figure 5. The testing layout in Figure 5 is designed to use the sensorization in the LARMbot torso V1 design within a proper testing frame. In particular, the IMU sensor and current sensor are those that are included in the control system, while the force sensors are installed at the top connection of the cables. This sensorization for testing has been arranged with the aim also not to affect the typical operation of the torso while extracting performance data in terms of motion characteristics and force transmission. Thus, the sensors are included in the torso design whose motion is given by servo motors that are controlled through an Arduino Mega by PC or a joystick.

Significant parameters are acquired during testing to evaluate the performance of the LARMbot torso V1. Based on the proposed setup, at LARM2 a testing layout is designed and implemented as shown in Figure 6a referring to the location of the used sensors.

Four force sensors (F_c1_ & F_c2_; F_c3_ & F_c4_) are located at suitable positions for measuring the forces between the cables and shoulder. Figure 6b illustrates the distribution of force sensors on the shoulder body. The type of the force sensor is RP-C7.6-LT model, [24], which is flexible ultra-thin, has ultra-low power consumption, extreme speed response, and high stability. The pressure induction range is 20 g ~ 1.5 kg. The activation time is less than 0.01 s. The response time of it is less than 10 ms. An IMU sensor, [25], is placed on the center of the shoulder body, which is indicated as point c in Figure 6a. The current sensor f in Figure 6a is used to calculate the power changes during the testing. The type of the current sensor is ACS 712, [26], which can measure the current from −5 A to +5 A. The PCA9685 board, [27], in e position in Figure 6a is used to connect driving servomotors the Arduino board in g. The Arduino board is also used to collect and manage the acquired data from the sensors to a laptop.

For operating the LARMbot torso V1 efficiently, a joystick is used as indicated with marker j in Figure 6a. LARMbot torso V1 can perform human-like motion in three modes following the trajectories in Table 2. By operating the joystick up and down, as related to the trajectory in Figure 7a, the LARMbot torso V1 can move backward and forward. When a user operates the joystick to move left and right, the torso can bend left and right, which is shown in Figure 7b. Moreover, the LARMbot V1 can move with a full rotation by combining the former two modes continuously, as shown in Figure 7c. According to previous experiments in [18,19,23], the LARMbot torso V1 can have a good performance on human-like motions, including bending left and right, and bending forward and backward as mode 1 and mode 2, respectively, as listed in Table 2.

Full rotation testing can validate the flexibility and the working performance of the humanoid torso in a more comprehensive way. In addition, the characterization of a full rotation of LARMbot torso V1 can give an overall evaluation of the torso capabilities figuring out problems to be considered for further improvements. For testing full rotation performance, an experiment is conducted as in the third mode of Figure 7c with data summarized in Table 2 whose parameters are monitored by using sensors. The joystick is operated in a counterclockwise direction for realizing the full rotation of the LARMbot torso V1. During the tests sensors installed on LARMbot torso V1 acquired data of the angular acceleration ω_H_ (x, y, z axis), linear acceleration a_H_ (x, y, z axis), rotation angles in pitch and roll, contact forces F_c1_ & F_c2_; F_c3_ & F_c4_ between the shoulder and cables, and the power consumption of the system. The expected motion characteristics are indicated in Table 2 yet as to fulfill the design requirements.

## 5. Results and Discussion

Tests with the LARMbot torso V1 were carried out in the three modes mentioned in Figure 7, and the LARMbot torso V1 moves as expected. A variety of human-like movements have been tested and reported in previous works, [17,18,23], with movements in flexion-extension, and lateral bending. The full rotation as the third mode was experienced in this work and results are reported in this paper only. The full rotation test is run with the planning in Figure 7c with the LARMbot torso V1 that first bends forward, then it starts to complete the rotating motion in the counterclockwise direction. Finally, the torso moves back to the home position and the servomotors stop. Test results are reported in Figure 8, Figure 9, Figure 10, Figure 11, Figure 12 and Figure 13 as illustrative of the torso capabilities with results from a representative test among the three repeated ones.

Figure 8 shows that the angular velocities of shoulder body as acquired by the IMU at point H change a lot during the full rotation test since the start at around 2.7 s with a proper time evolution. The angular velocity ω_x_ starts to decrease and at about 3.2 s reaches a minimum value of about −14 deg/s whereas the maximum is detected of about 11 deg/s. Then it decreases again until about 4.7 s. Finally, the curve increases to 0 deg/s at the rest position. The angular velocity ω_y_ shows the same trend of ω_x_ but reaching to maximum and minimum values earlier than ω_x_. When the time is about 2.6 s, ω_y_ reaches the first maximum value of around 13 deg/s. Then, at about 3.5 s, the maximum value of ω_y_ is about 14 deg/s. When the time is around 4.5 s, the third maximum value is detected of about 10 deg/s, from which ω_y_ decreases to about 7 deg/s after the time is about 5 s. The values of ω_x_ and ω_y_ change several times accordingly to the sequence of bending movements. The value of ω_z_ does not change significantly during the tested full rotation motion since the full rotation does not include twisting of the torso. As the time near 2.5 s, ω_z_ increases until around 4.7 s when it decreases to about 1 deg/s. The maximum values of angular velocities are less than 15 deg/s well reflecting the smooth operation of the full rotation mode with the expected cyclic time evolution.

The acquired linear acceleration components of shoulder point H are reported in Figure 9. The linear acceleration a_Hz_ does not change significantly form the sensed gravity acceleration. On the contrary, a_Hx_ and a_Hy_ show significant variations. When the time is about 2.3 s, a_Hx_ starts to decrease and reaches the minimum value of about −1.8 m/s^2^ when the time is about at 3 s. After that, it increases and stays at a constant level after 4.5 s a_Hy_ increases when the time is about 2.5 s and at about 3.5 s reaches a maximum of about −0.3 m/s^2^. Then, it decreases to −1.9 m/s^2^. And after that shows a small increase up to a quasi-constant value. The time evolution with those limited acceleration values well describes the time evolution during the full rotation torso motion.

Figure 10 shows that the time evolution of the angles in pitch and roll with significant variations well corresponding to the previous results reflecting the feasible characteristics of human-like motion in the torso full rotation. The roll angle starts to increase when the time is about 2.3 s. When the time is about 3.3 s, it increases to maximum of about 4 deg. After that, it decreases until the time is about 4.5 s and then, a last variation from about −3 deg to about −0.5 deg. Finally, it does not change significantly after 5 s. The pitch angle starts to decrease when the time is about at 2.3 s and at about 3 s it reaches the maximum value of about −8 deg at the opposite direction. After that, it increases until about 4 s. and finally, it decreases from 1 deg to about −1 deg that remains quasi constant.

The computed power of servomotors is reported in Figure 11. The power starts to increase at about 2 s when the servomotors start to work. When the time is about 2.7 s, it reaches the maximum of about 5.8 W. Then, it decreases and reaches a new maximum of about 6.5 W when the time is about 4.2 s. After 5 s, the power remains at a constant value of about 1 W. During the torso full rotation, the four servomotors work together most of the time. The values of power are larger than in bending left and right or forward and backward and therefore, completing the full rotation motion requires more power.

Figure 12 shows the measured forces F_c1_ and F_c2_ between the cables and shoulder body. F_c1_ starts to increase when the time is about 2.3 s at the same time that F_c2_ decreases as per the antagonist action. When the time is about 2.5 s, F_c1_ reaches maximum value of about 1.1 N while the cable C_2_ loses tension with F_c2_ going to 0 N. Around 3 s, F_c1_ starts to decrease, and t F_c2_ changes in the opposite way. When the time is about 4.5 s, a similar situation happens again. After 5 s, F_c1_ and F_c2_ remain at a constant level of about 0.2 N and 1.0 N, respectively.

Figure 13 shows the time evolution of the contact forces between the cables C_3_ and C_4_, and the shoulder body at the corresponding connection points. At about 2.5 s, F_c3_ starts to increase and at the same time F_c4_ decreases as per the antagonist action. When the time is about 3.3 s, F_c4_ reaches a maximum of about 1.1 N but at about 3 s F_c4_ goes to 0 N indicating that e cable C_4_ loses tension for a while. When the time is about 4 s, the F_c3_ is 0 N while F_c4_ reaches the maximum of about 1.2 N. After that, F_c3_ increases and F_c4_ decreases. When the time is about 5 s, both of tensions start to remain at a constant level of 0.5 N and 0.45 N, respectively.

The full rotation results reveal a time evolution with smooth regular variation, indicating that the LARMbot torso V1 can successfully mimic smooth human-like torso rotation motion. In summary, because the LARMbot torso V1 rotated at a relatively fast speed, the angular accelerations in the x and y directions changed abruptly at certain moments. However, linear accelerations did not show large changes. During operation in mode 3, the LARMbot torso V1 moved forward at the beginning with the pitch and roll angles changing drastically. After that, the angles changed more regularly. The contact forces between the shoulder and cables were less than 1.3 N, i.e., considerably less than the admissible stress of the PLA material used in the torso structure. In addition, the metal cables were sufficiently stiff to bear the force. It is worth noting that the cables lost tension at times, i.e., one cable was too short and the other was too long. To drive the whole system, the required power was determined to be less than 7 W. These observations show that the LARMbot torso V1 can mimic human-like motions efficiently and can satisfy the requirements of a torso for a humanoid robot. Nonetheless, deficiencies that will require improvements, both in design and operation aspects, were found.

The prototype of LARMbot torso V1 can complete full rotations continuously, without any sudden changes or stops. As such, the acquired data did not show changes during the tests, as illustrate in the plots in Figure 8, Figure 9, Figure 10, Figure 11, Figure 12 and Figure 13. This behavior can be considered satisfactory to reproduce a smooth human-like motion whereas a smoothness motion characteristic can be defined as related to the capability to move continuously with regular movements with no significant variations.

## 6. Considerations for a New Design

During our experiments, the LARMbot torso V1 showed smooth full-rotation motion that was well suited for human-like motions in the assigned tasks. However, problems were identified, for which solutions will now be suggested as in Figure 14:Motion limitation. The LARMbot torso V1 can move with limited human-like motions compared with humans. Hence, it is imperative to enlarge the motion workspace of the humanoid torso, i.e., with larger bending angles. In addition, the joint of the humanoid torso should be designed and built with proper stiffness in order to rotate in different directions.Limited payload. The LARMbot torso V1 supported arms that were capable of operating under a limited payload. Therefore, the new design should seek to accommodate humanoid arms with proper large payload capacities. In addition, revising the vertebral-like spine structure would also be useful for better stiffness performance under high payloads.Cost and power consumption. Four servomotors require considerable actuation power to drive the torso system, which also increased the cost of the prototype. Therefore, using fewer servomotors would be a convenient way to obtain a more efficient structure and control system by saving on equipment costs and power consumption.Motion control system. The current open-loop control system could control the LARMbot torso V1 satisfactorily. However, sometimes it was found to be ineffective in terms of driving the servomotors. Hence, a closed-loop control system should be used for more efficient performance in autonomous and supervised modes.Operation efficiency. The cables occasionally lost tension in antagonist mode, causing problems in terms of motion accuracy and control, as well as reducing energy efficiency. Designing an automatic operation mode could improve the human-like motions during autonomous operations. In addition, a voice controller may be convenient for users.

New vertebra-disc unit designs could be considered based on the problems observed with the LARMbot torso V1. Based on the above suggestions, two new designs for a vertebral-disc unit are proposed, as shown in Figure 15.

The first, with a torsion spring at G_c_, is shown in Figure 15a. It consists of two vertebras, two cables, a flexural spring, a drum with a torsion spring, and a servomotor. The A_1_A_2_ is the fixed vertebra with the length L_f_. The cables with length l_c1_ and l_c__2_ go through the fixed vertebra and connect with drum G_c_ and servomotor, respectively. The moving vertebra B_1_B_2_ with length L_m_ can perform motions via cooperation with the servomotor and the drum with torsion spring. The ϕ angle of the moving vertebra shows the relative motion between two vertebras. The tension forces on cables are indicated as T_c1_ and T_c2_. The vertebra-disc unit can complete motion in several steps. For example, when the servomotor rotates 90 deg, the unit is at the home position; when the servomotor rotates 0 deg, the moving vertebra bends right; and when the servomotor rotates 180 deg, the moving vertebra bends left. During this operation, a drum with a torsion spring will rotate with a proper balancing force to keep the cable under tension. To obtain the desired functionality, the torsion spring should be sized appropriately.

The second design with a tension spring is shown in Figure 15b. It comprises two vertebras, a servomotor, a flexural spring, and a tension spring. A cable goes through the fixed vertebra A_1_A_2_ with length L_f_. The length of the cable is l_c2_ with tension force T_c2_. The tension spring is characterized by length *l**_SC_* and spring coefficient K_SC_, and the tension on it is indicated by T_s_, given by T_c1_. The servomotor rotates, thereby changing the length of the cable that is wrapped around a drum, and the force and the length of the tension spring change accordingly. The upper vertebra moves with the proper angle in a motion relative to the fixed one. During this operation, the servomotor rotates in the same way as that in the first design, and the moving vertebra bends in a similar way.

These two conceptual additions can achieve the desired motion and satisfy the design requirements, thereby solving the problems described in Figure 14. The new designs also decrease the number of servomotors required by the mechanism, which would reduce costs and power consumption. In addition, springs are a good way to store energy for the vertebra-disc unit, decreasing the power required to drive the actuating cables. However, there may be new problems in the new designs. In terms of the first design, a torsion spring should be selected with proper stiffness so that the vertebra moves correctly; additionally, it should be designed with a proper size so that it can be arranged with a small drum in a limited space. As for the second design, the length and the tension on the tension spring should be designed so that a stationary force can be supplied to the moving vertebra. In addition, connecting several vertebra-disc units may also remain a challenge.

## 7. Conclusions

In this paper, the human-like motion of the LARMbot torso V1 was tested, including a full-rotation experiment. The mechanical performance torso was analyzed to experimentally characterize the operation of the current prototype. A control system was designed using an Arduino Mega board for further tests. Performance characteristics were obtained via sensors for angular velocity, linear acceleration, motion angles, forces between the cables and shoulder, and power consumption. The LARMbot torso V1 showed satisfactory performance of human-like motions, i.e., the rotation angles did not undergo sudden changes and yielded the proper values, the contact forces at the cable connections were less than 1.3 N, and the required power was less than 7 W. Nonetheless, suggestions for improvements were made for two possible new designs. The contributions of this paper are as follows:To update the control system and perform a smooth, full rotation of the LARMbot torso V1.To analyze the relevant characteristics of the human torso and propose requirements for a new design of the LARMbot torso.To test the LARMbot torso V1 in order to obtain experimental results with which to can better characterize the current porotype, including its shortcomings.To propose new designs of vertebral-disc units, considering the test results for the current prototype version.

## Figures and Tables

**Figure 1 micromachines-13-01548-f001:**
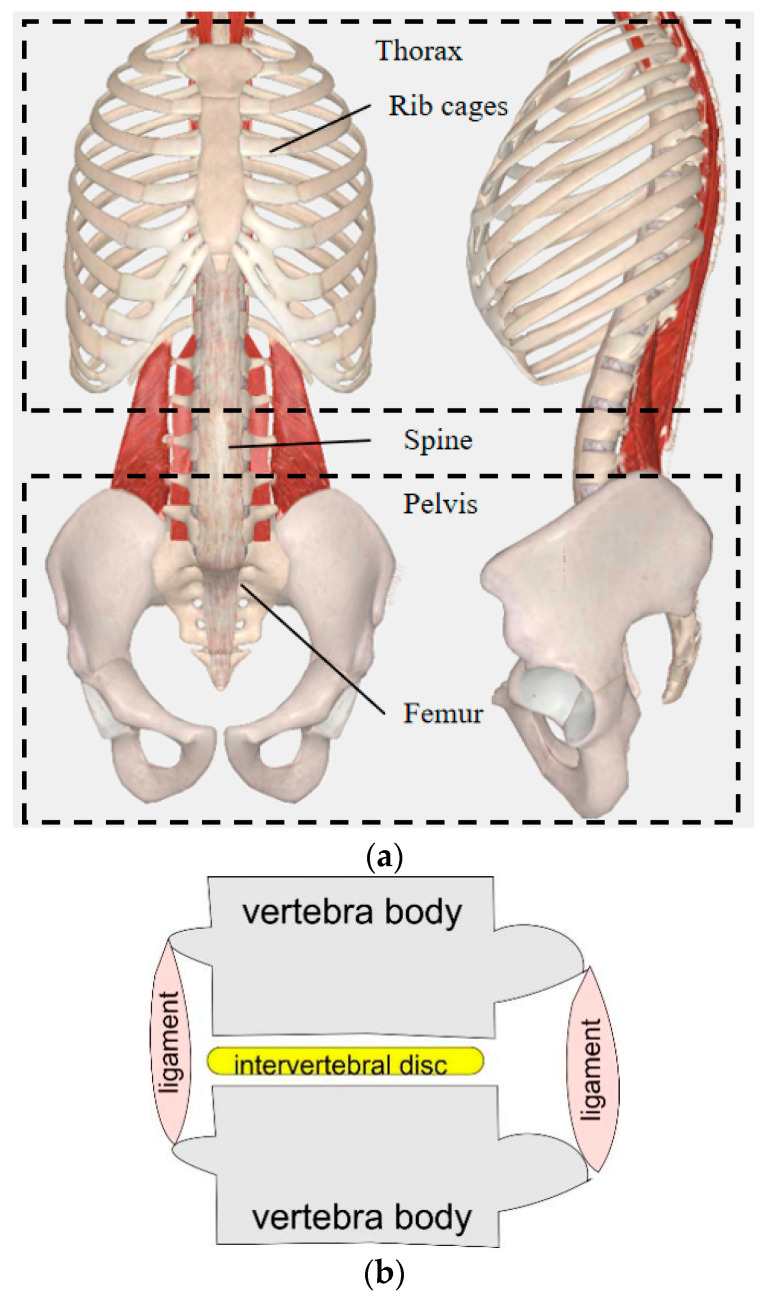
Schematics of the human torso: (**a**) skeleton structure; (**b**) the spine unit.

**Figure 2 micromachines-13-01548-f002:**
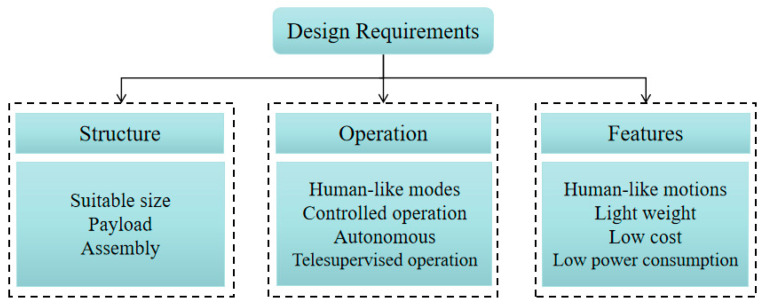
Design requirements of a humanoid torso.

**Figure 3 micromachines-13-01548-f003:**
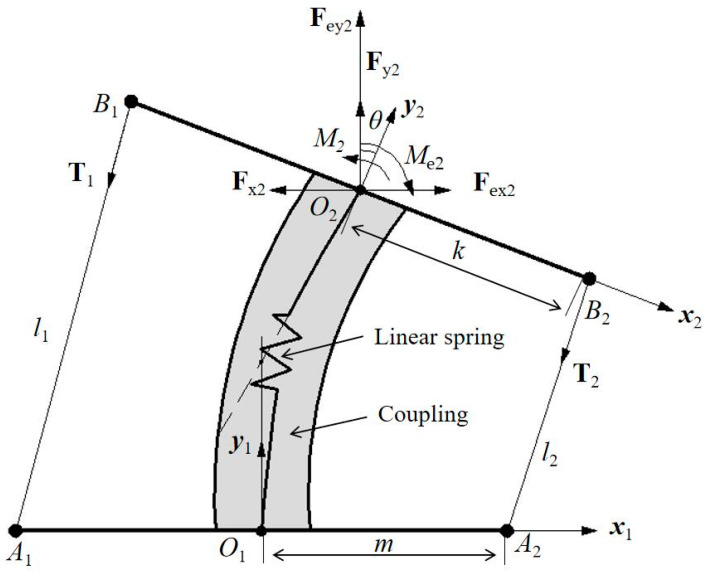
A model of cable-driven mechanism for the vertebra-disc unit with parameters.

**Figure 4 micromachines-13-01548-f004:**
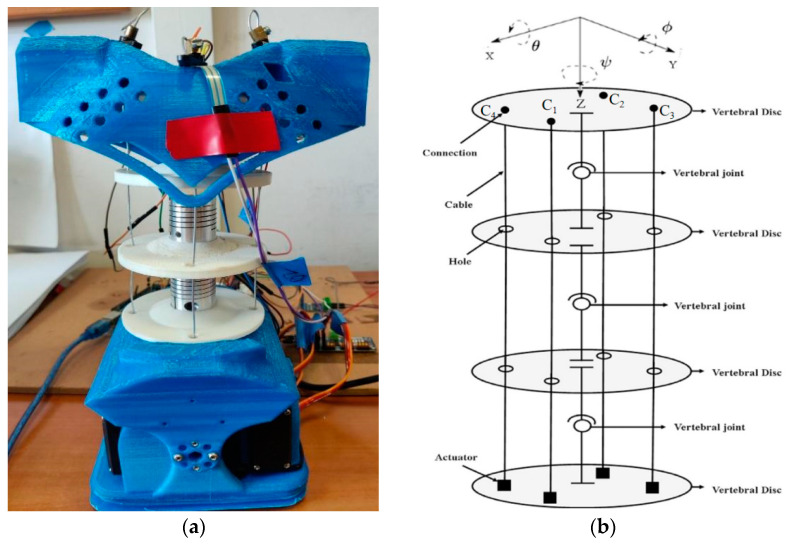
LARMbot torso V1 at LARM2 laboratory in Rome: (**a**) a prototype; (**b**) a scheme [23].

**Figure 5 micromachines-13-01548-f005:**
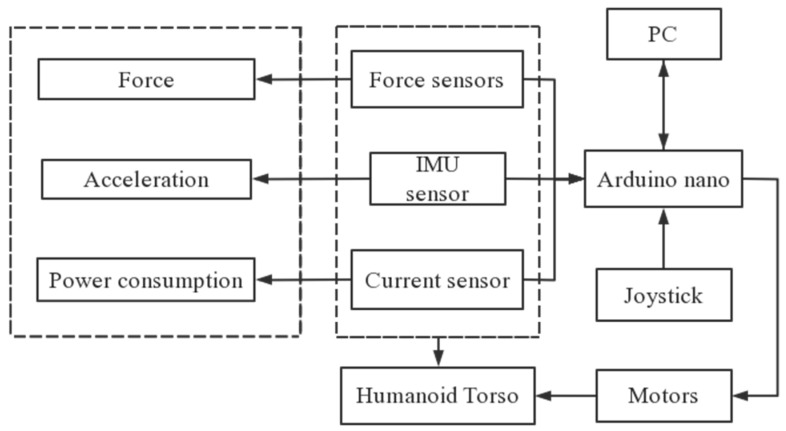
Testing layout of LARMbot torso V1.

**Figure 6 micromachines-13-01548-f006:**
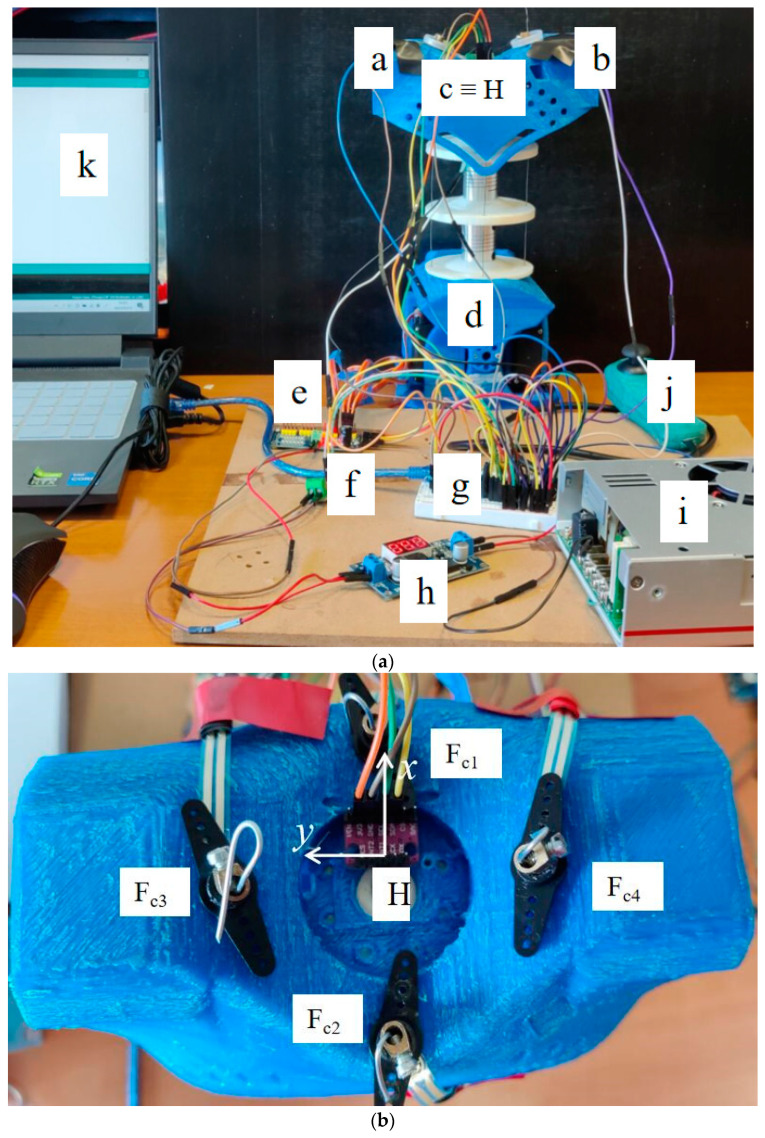
Distribution of sensors in the testing lab setup: (**a**) overall; (**b**) on shoulder body. (a: force sensor F_c4_; b: force sensor F_c3_; c: IMU sensor; d: LARMbot torso V1; e: PCA9685; f: current sensor; g: Arduino nano; h: DC-DC convert; i: power supply; j: joystick; k: laptop).

**Figure 7 micromachines-13-01548-f007:**
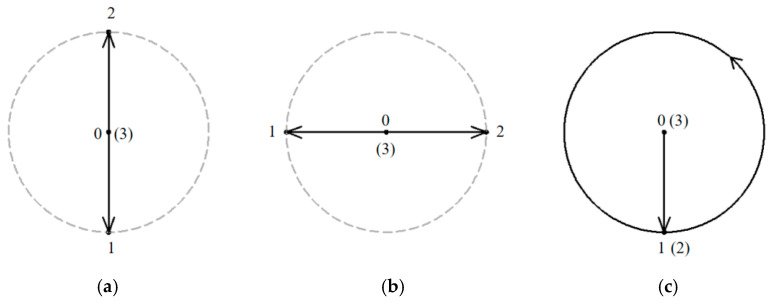
Schemes for joystick-controlled motions of the LARMbot torso: (**a**) bending forward and backward; (**b**) bending left and right; (**c**) full rotation.

**Figure 8 micromachines-13-01548-f008:**
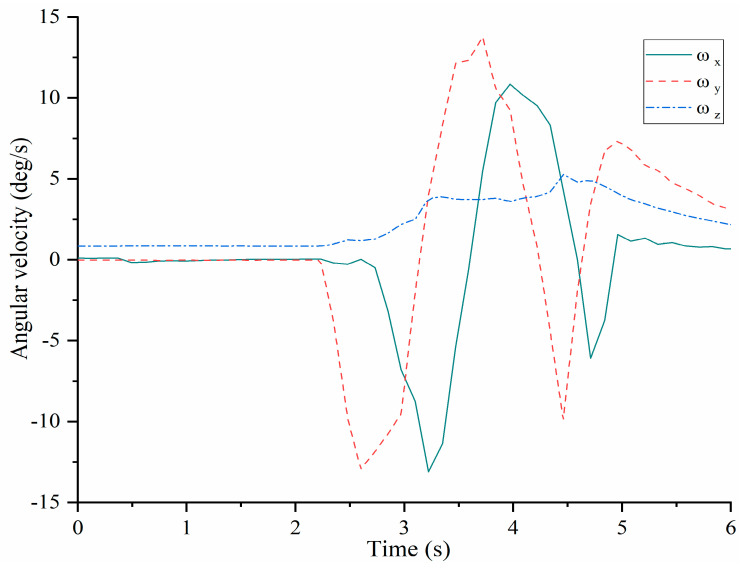
Acquired angular velocity of the shoulder during a full rotation testing, Table 2.

**Figure 9 micromachines-13-01548-f009:**
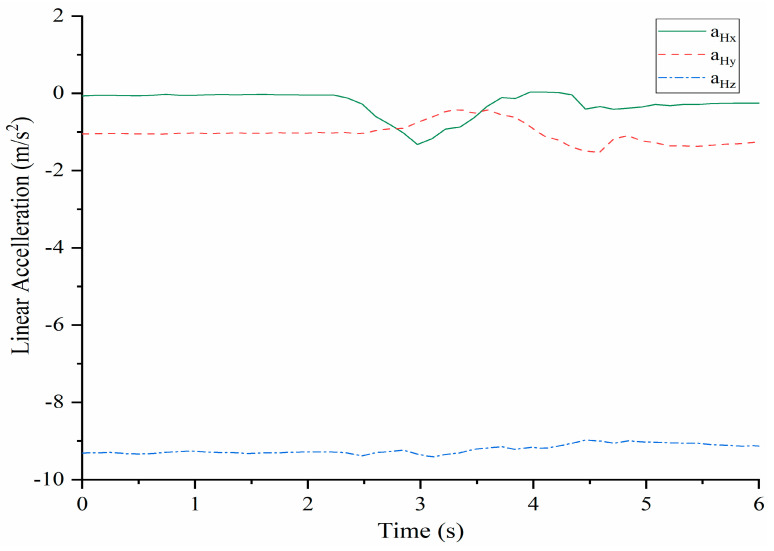
Acquired linear acceleration of the shoulder point H during full rotation testing, Table 2.

**Figure 10 micromachines-13-01548-f010:**
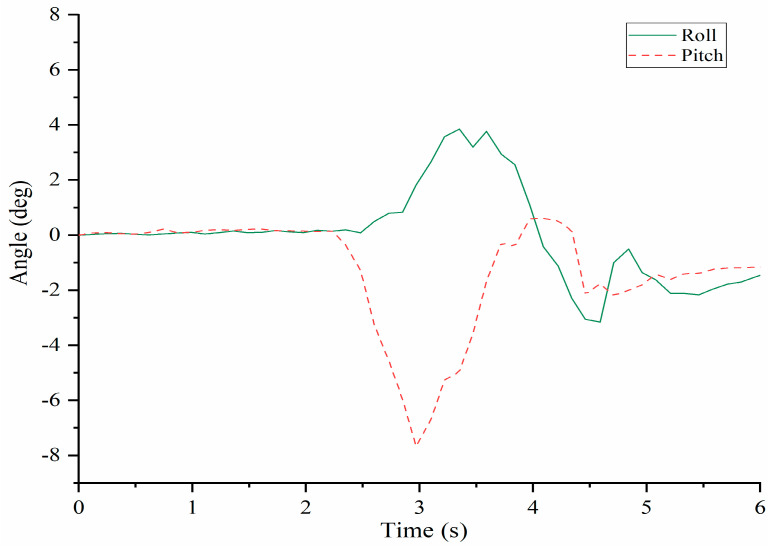
Rolland pitch angles of the shoulder body during full rotation, Table 2.

**Figure 11 micromachines-13-01548-f011:**
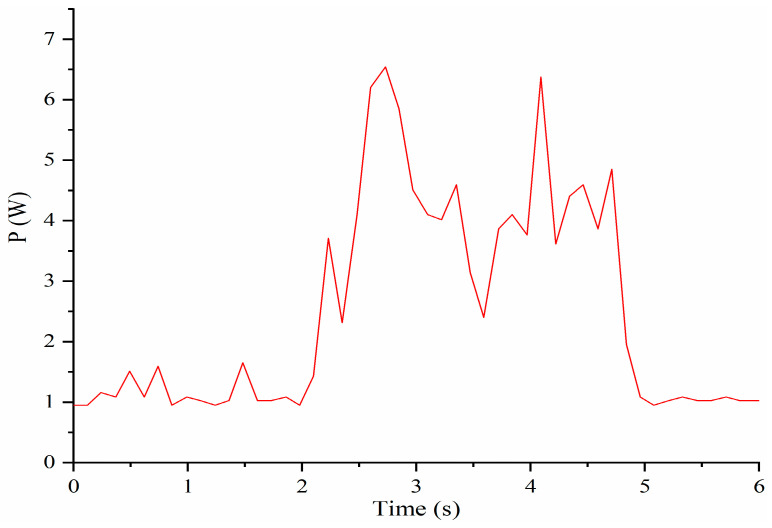
Computed power consumption of servomotors during full rotation testing, Table 2.

**Figure 12 micromachines-13-01548-f012:**
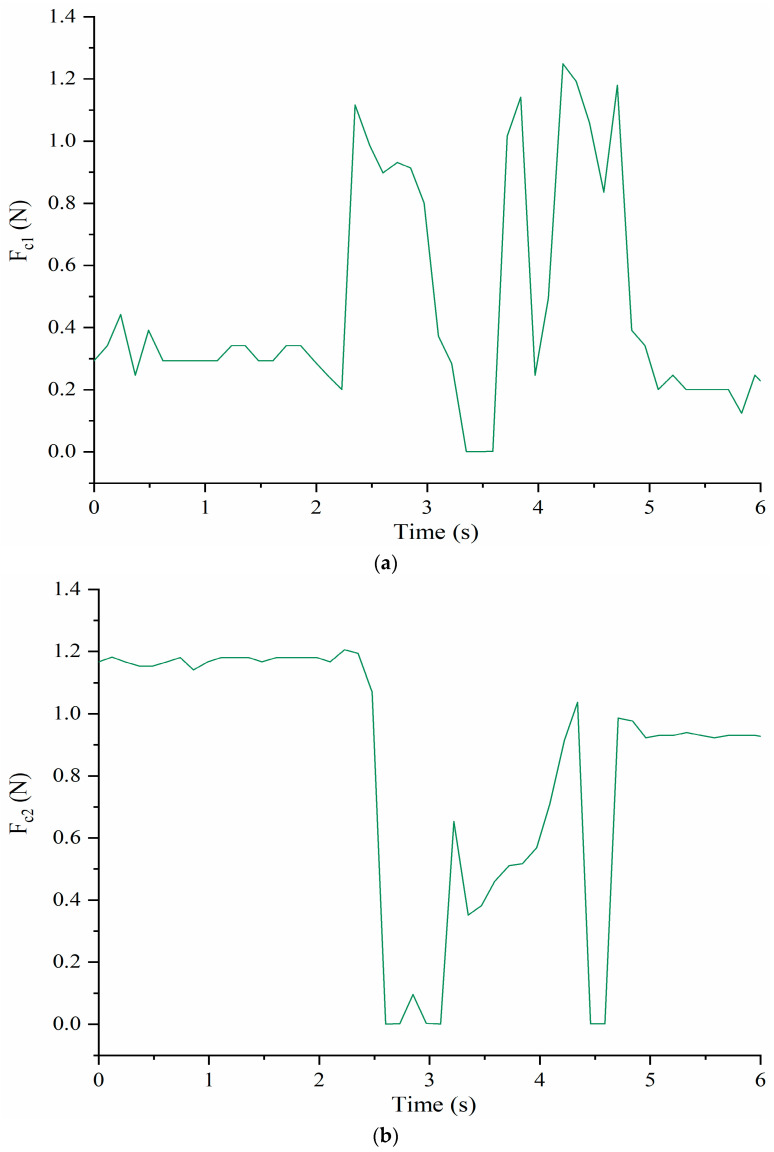
Acquired forces between the cables and shoulder body during full rotation: (**a**) F_c1_; (**b**) F_c2_.

**Figure 13 micromachines-13-01548-f013:**
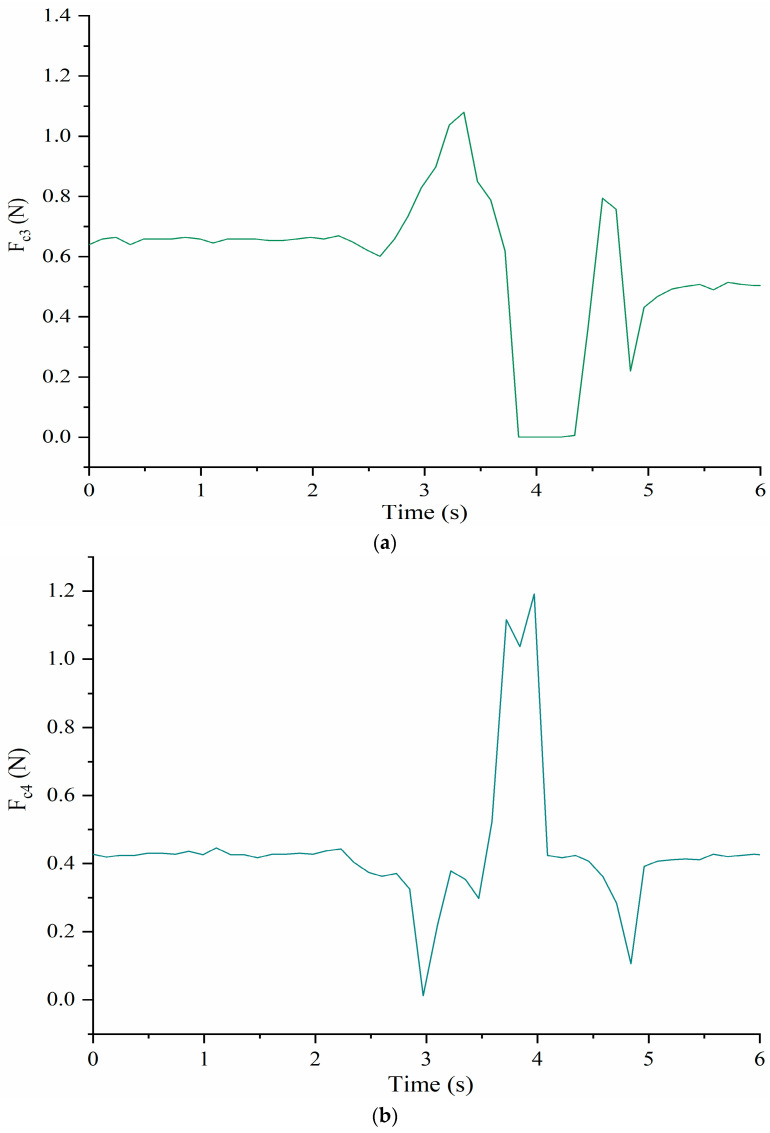
Acquired forces between the cables and shoulder during a full rotation: (**a**) F_c3_; (**b**) F_c4_.

**Figure 14 micromachines-13-01548-f014:**
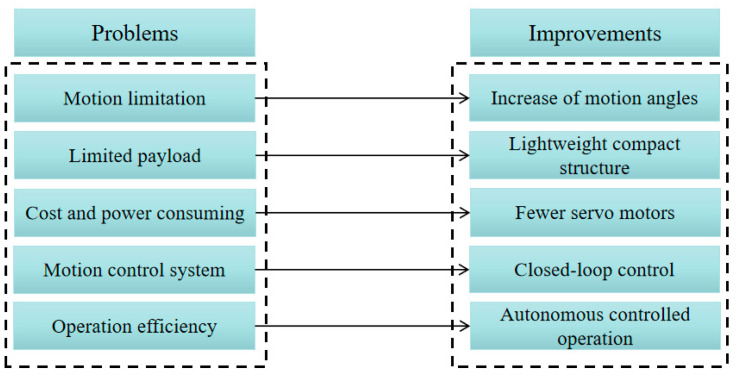
Problems and improvements for a new design of LARMbot torso.

**Figure 15 micromachines-13-01548-f015:**
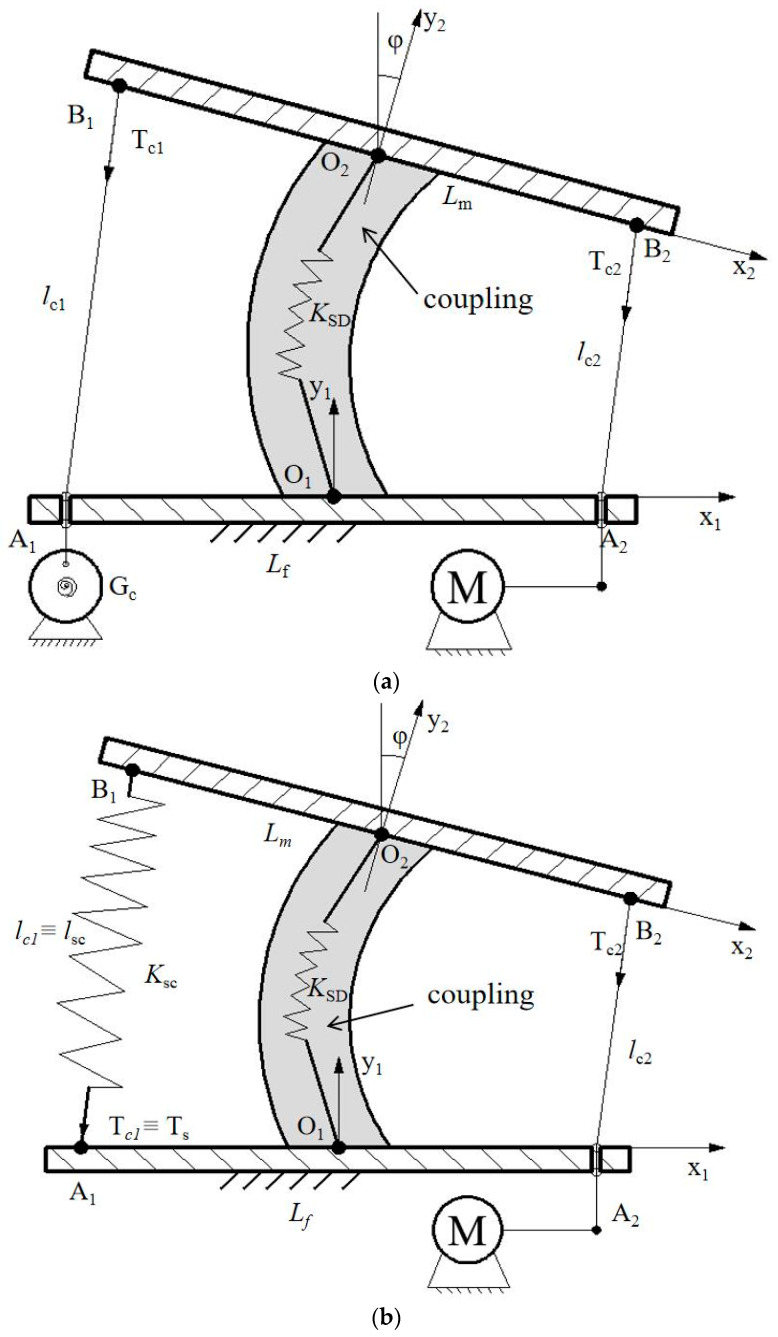
Conceptual design for a new vertebra-disc unit: (**a**) with torsion spring; (**b**) with linear spring.

**Table 1 micromachines-13-01548-t001:** Motion ranges of human torso in three modes [17,20].

Motions	Flexion Extension		Lateral Bending		Transverse Rotation	
Figures	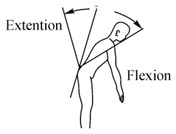		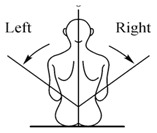		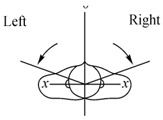	
Angles	Front: 45 deg	Back: 30 deg	Left: 40 deg	Right: 40 deg	Left: 50 deg	Right: 50 deg

**Table 2 micromachines-13-01548-t002:** Testing modes for experimental performance characterization of LARMbot torso V1.

No.	Modes	Range of Motions (deg)	Range of Cable Force	Range of Expected Power Consumption
1	Bending forward and backward	−13~10	<2 N	<3 W
2	Bending left and right	−15~15	<2 N	<3 W
3	Full rotation	−5~5	<2 N	<8 W

## Data Availability

The data used to support the findings of the study are available from the corresponding author upon request.

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
