# Peer review of "Design and Performance Analysis of LARMbot Torso V1"

_micromachines, 2022, doi:10.3390/mi13091548_

Round 1

Reviewer 1 Report

The manuscript reports on a study devoted to a design and performance analysis of a particular humanoid torso, i.e., the LARMbot torso V1, which has previously been introduced and discussed in [18, 19, 23]). In terms of topic, the manuscript is suited to the Journal. However, with respect to novelty, experimental settings and result presentation, the manuscript has some serious flaws. When interpreting the following remarks please note that I believe that previous work on the LARMbot torso V1 is valuable in itself, and that the remarks relate only to the considered manuscript.

Remarks:

Novelty:

1. Keeping in mind the previous publications of the second author (cf. [18, 19, 23], all adequately referenced in the manuscript), the manuscript lacks sufficient novelty. The considered humanoid torso has been introduced and partially evaluated in [18, 19, 23] and the vertebra-spring cable-driven mechanism (and accompanying equations (1)-(7)) has already been elaborated in [22] (and adequately cited by the authors). Thus, the planned contribution of the manuscript was aimed at additional experimental evaluation of the torso. However, although the authors announce that they test the torso “in a wider operation mode” (p. 1, l. 68-69) and “in a more complete way” (p. 9, l. 247), they actually just report the measurement obtained from a set of sensors during the full torso rotation. This is not a sufficient contribution, especially since the reported results are not discussed with respect to any external criteria intended to evaluate the human-likeness of the torso movement, which was also announced (e.g., cf. p. 9,  l. 262-264). Please cf. also the following two remarks.

Experimental settings:

2. Related to the reported settings, the authors briefly explain the deployment of sensors, the observed parameters, how the joystick can be operated in order to control the servo motors, and state that the “experiments are conducted for the three modes” (p. 10, l. 260-261): bending forward and backward, bending left and right, and full rotation (cf. Fig. 7). However, the following important questions remain unclear:

2.1 The authors use words “experiment” and “experimental” extensively throughout the manuscript, but is remain unclear what is actually the “experimental component” in their settings? The experimental protocol is completely omitted. What is the control variable? Who was the operator? Was the parameter measurement repeated and how many times? From the provided description, it appears that the authors conducted a measurement rather than an experiment.

2.2 The authors report only the results obtained for the third mode (i.e., full rotation), as “emblematic illustrative example” (p. 10, l. 265). Why is the characterization of a full rotation of the torso so “emblematic and illustrative” in the observed context?

Results:

3. The obtained results are just descriptively reported in Section 5. However, the stated conclusion that “The results of a full rotation test show that the LARMbot torso V1 can success fully mimic human-like motion with smoothness” (p. 14, l. 348-349) is not supported. Furthermore, the criteria to evaluate the “smoothness” of the torso movements were not introduced (neither qualitatively nor quantitatively).

Additional remarks:

4. The point of departure for this manuscript (cf. Section 1) is that the existing humanoid torsos suffer from certain disadvantages, e.g., they are not lightweight enough (p.1, l. 33), their structures are complicated and not of high modularity (p.2, l. 50-51). Although these statements are true, why are they important in the context of the reported study?

5. Abbreviation IMU (e.g., inertial measurement unit) should have been defined on its first occurrence in the manuscript (p. 2, l. 72).

6. The authors use several related but distinct notions: “human-like assignments” (e.g., cf. p. 1, l. 23), “human-like motions” (e.g., cf. p. 2, l. 81), and “human-like modes” (cf. Fig. 2). If all of these notions are necessary, the authors should clearly define and carefully use them in order to avoid the conceptual confusion about the objective of the study.

7. Fig. 1b (e.g., a spine model) needs some clarification.

8. The following sentence is not clear in the given context: “The mechanical design of the investigated humanoid torso as well its operation and control are sized at mini size level with biomimetic features of biological characteristics and functions for humanoid robots” (p. 2, l. 69-71).

9. The authors refer to Fig. 2b (p. 4, l. 135). However, there is not such a figure in the manuscript.

10. Certain elements in Fig. 6a are denoted with letters “a”, “b”, …, “k” (“j” is missing), but the authors do not refer to these marking. Why are they needed in Fig. 6?

11. In Section 2, the authors discuss certain design requirements for a humanoid torso, although most of them are not considered in the reported study.

Author Response

see attacehd file

Reviewer 2 Report

Paper in subject of Micromachines journal of MDPI

Paper present design and analysis of new solution of torso of humanoid robot developed by authors.

Pap

Comment 1

Paper based on adequate references strickle connected with title. I propose add some articles in scientific background of article about bipedal robots especially humanoid type

Sensors 202222, 4440. https://doi.org/10.3390/s22124440

Curr Robot Rep 2021, 2, 201–210 (2021). https://doi.org/10.1007/s43154-021-00050-9

Comment 2

Please consider to add at the end of Introduction the description of paper structure

Comment 3

You present Figure 1a and Figure 1b. The copyrights of pictures are obtained?

It needs to check copyright problem for all figures presented in article

Comment 4

In line 139  you wrote

“The intervertebral disc is modeled as a flexible coupling with a spring. The mechanism is driven by 2 servo motors at points A 1 and A 2 in Figure 3.”

Can you add to Figure 3 the servo motors in points: A1 and A2

Comment 5

You wrote

4. Testing setup and modes

Because you describe of testing setup and plan of experiment please consider to changing it on

4. Materials and methods

and use of subsections

Comment 5

You wrote

5. Results

Because you both present results of experiment and discuse it I propose change on

5. Results and discussion

Please consider to add some subsections to clear your description

Comment 6

You wrote in Section 6

However, according to the reported testing results, problems can be considered for improvements that can be suggested as summarized, Figure 14:

A. Motion limitation. The LARMbot torso V1 can move with limited human-like space compared with human motions. Hence, it is imperative to increase work….

Because A., B.,C… not used in Figure 14 I propose to use of bullet

However, according to the reported testing results, problems can be considered for improvements that can be suggested as summarized, Figure 14:

·         Motion limitation. The LARMbot torso V1 can move with limited human-like space compared with human motions. Hence, it is imperative to increase work….

Comment 7

In Conclussion I propose use bullet to beter indicate your achievements in paper

Author Response

see attacehd file

Round 2

Reviewer 1 Report

The authors have responded to my remarks from the previous review report and I believe that the manuscript has been sufficiently improved to warrant publication in Micromachines.

Remark:

The formulation of one of the stated contributions, i.e., "To analyze the characteristics of the human torso" (p. 18, l .525), appears too general. The authors are suggested to consider the possibility to reformulate this statement  in order to make it more specific. E.g., "selected characteristics of human torso", "relevant characteristics of human torso", etc.

Author Response

see attahed file
